# Karyotype Evolution in 10 Pinniped Species: Variability of Heterochromatin versus High Conservatism of Euchromatin as Revealed by Comparative Molecular Cytogenetics

**DOI:** 10.3390/genes11121485

**Published:** 2020-12-10

**Authors:** Violetta R. Beklemisheva, Polina L. Perelman, Natalya A. Lemskaya, Anastasia A. Proskuryakova, Natalya A. Serdyukova, Vladimir N. Burkanov, Maksim B. Gorshunov, Oliver Ryder, Mary Thompson, Gina Lento, Stephen J. O’Brien, Alexander S. Graphodatsky

**Affiliations:** 1Department of Comparative Genomics, Institute of Molecular and Cellular Biology, Siberian Branch of Russian Academy of Sciences, 630090 Novosibirsk, Russia; polina.perelman@gmail.com (P.L.P.); lemnat@mcb.nsc.ru (N.A.L.); andrena@mcb.nsc.ru (A.A.P.); ns3032@yandex.ru (N.A.S.); graf@mcb.nsc.ru (A.S.G.); 2Department of Higher Vertebrate Ecology, Kamchatka Branch of Pacific Geographical Institute of Far East Branch of Russian Academy of Sciences, 683000 Petropavlovsk-Kamchatsky, Russia; Vladimir.Burkanov@noaa.gov; 3Institute of Biological Problems of the North, Far East Branch of Russian Academy of Sciences, 685000 Magadan, Russia; mbgmmg@mail.ru; 4San Diego Zoo Global, San Diego, CA 92101, USA; ORyder@sandiegozoo.org; 5BSP-CCR Genetics Core, Center for Cancer Research, National Cancer Institute, 8560 Progress Drive, Frederick, MD 21702, USA; thompsonm@mail.nih.gov; 6ShanLen Enterprises, Austin, TX 78757, USA; ginalento@gmail.com; 7Computer Technologies Laboratory, ITMO University, 49 Kronverkskiy Pr., 197101 St. Petersburg, Russia; lgdchief@gmail.com; 8Guy Harvey Oceanographic Center, Nova Southeastern University Ft. Lauderdale, 8000 North Ocean Drive, Ft. Lauderdale, FL 33004, USA

**Keywords:** fluorescence in situ hybridization, telomere repeat, rDNA probe, CBG staining, CDAG banding, constitutive heterochromatin, pericentric inversion, semiaquatic mammal, seal, walrus, tandem fusion, evolutionary new centromere, chromosome map

## Abstract

Pinnipedia karyotype evolution was studied here using human, domestic dog, and stone marten whole-chromosome painting probes to obtain comparative chromosome maps among species of Odobenidae (*Odobenus rosmarus*), Phocidae (*Phoca vitulina*, *Phoca largha*, *Phoca hispida*, *Pusa sibirica*, *Erignathus barbatus*), and Otariidae (*Eumetopias jubatus*, *Callorhinus ursinus*, *Phocarctos hookeri*, and *Arctocephalus forsteri*). Structural and functional chromosomal features were assessed with telomere repeat and ribosomal-DNA probes and by CBG (C-bands revealed by barium hydroxide treatment followed by Giemsa staining) and CDAG (Chromomycin A3-DAPI after G-banding) methods. We demonstrated diversity of heterochromatin among pinniped karyotypes in terms of localization, size, and nucleotide composition. For the first time, an intrachromosomal rearrangement common for Otariidae and Odobenidae was revealed. We postulate that the order of evolutionarily conserved segments in the analyzed pinnipeds is the same as the order proposed for the ancestral Carnivora karyotype (2n = 38). The evolution of conserved genomes of pinnipeds has been accompanied by few fusion events (less than one rearrangement per 10 million years) and by novel intrachromosomal changes including the emergence of new centromeres and pericentric inversion/centromere repositioning. The observed interspecific diversity of pinniped karyotypes driven by constitutive heterochromatin variation likely has played an important role in karyotype evolution of pinnipeds, thereby contributing to the differences of pinnipeds’ chromosome sets.

## 1. Introduction

Genome analysis for related species within taxonomic assemblages enriches our evolutionary perspective and provides new information about the speciation process. Comparative chromosome maps obtained by means of chromosome-painting probes reveal patterns of karyotype transformations during evolution and allow to formulate hypotheses about ancestral karyotype composition. Constitutive heterochromatin (CH), an important genome component with a cryptic role and poorly understood functions, is filled with diverse and often complex repetitive elements that are challenging for resolution and can confound bioinformatic assembly of genome sequences. Repeated sequences perform regulatory functions and ensure spatial organization of chromatin in the nucleus [1,2]. CH may be involved in structural chromosomal rearrangements [3] due to nonhomologous-recombination susceptibility [1]. Heterochromatin may be present in the genome as near-centromeric blocks, interstitial blocks, or additional chromosomal arms. CH changes may occur in conserved genomes and in highly rearranged genomes [4]. Comparative chromosome maps can help to unravel repeat structure and organization, to bridge scaffolds, and to validate chromosome level assemblies of genome sequences.

In recent decades, patterns of mammalian genome evolution were revealed by determining the rates of chromosome exchanges and by documenting the types of rearrangements within major orders that had shaped genomes of extant species on the basis of reconstructed ancestral karyotypes [5,6,7,8,9,10]. Carnivora (includes two branches: Feliformia and Caniformia) is an order with well-characterized patterns of chromosome evolution [11]. There are two types of evolutionary changes that dominate in certain groups of the order: chromosomal sets of Canidae, Ursidae, and Viverridae have derived mainly from the fusion–fission of ancestral elements, whereas inversions and centromere repositioning events have mostly occurred during karyotype evolution of other families of Carnivora.

Among Caniformia families, Pinnipedia (seals and walrus) has a nearly unchanged ancestral karyotype. Pinnipeds represent a group of semiaquatic animals with advanced adaptations to life in water, thus raising a hypothesis that their conserved genome disposition is associated with acquired aquatic adaptations.

Ulfur Arnason has revealed remarkable karyotype conservatism of this group by classic and banding cytogenetics [4,12]. This high syntenic conservatism has been confirmed by molecular cytogenetics [8,9], which has identified the minimum number of fusion events during 25 million years of pinniped radiation into three families: Odobenidae, Otariidae, and Phocidae. In the 1980s–1990s, molecular composition of heterochromatin in Carnivora was studied [13,14,15,16,17,18]. Different classes of repeats were shown to characterize different families, pointing to rapid evolution and repeat diversification during speciation. With only a few genomes of pinnipeds sequenced and assembled into chromosomes to date [19], comparative chromosome maps would be an important step in pinniped genome research.

Here we investigated remarkable karyotype conservatism among pinnipeds by chromosome painting in a wider range of Otariidae and Phocidae species. With the help of high-resolution canine probes, we tested for the intrachromosomal rearrangements that could be present in otherwise conserved syntenic groups. We determined whether heterochromatin variation plays some part in the genomes featuring high levels of syntenic conservatism.

## 2. Materials and Methods

### 2.1. Species Sampled and an Ethics Statement

Tissue samples from wild-caught animals were used here (Table 1). The collection of samples from the walrus, Steller sea lion, and Baikal seal has been described earlier [9]. Samples from the bearded seal and ringed seal were collected during aboriginal quota sealing in the coastal waters of the Bering Sea (Mechigmen Bay, Chukotka, Russia). Ear biopsy was performed on northern fur seals under inhalation anesthesia (isoflurane) during a veterinary examination. Tissue biopsies of the spotted seal were obtained from the animals sampled for research purposes by the Magadan branch staff of the Russian Research Institute of Fisheries and Oceanography. All of the tissue samples were collected according to procedures approved by the Ethics Committee on Animal and Human Research at the Institute of Molecular and Cellular Biology, Russia (protocol No. 01/20 of 11 February 2020). The harbor seal, New Zealand fur seal, and New Zealand sea lion fibroblast cultures were obtained from the Laboratory of Genomic Diversity (National Cancer Institute, Frederick, MD, USA).

### 2.2. Cell Culture and Chromosome Preparation

Storage and transportation of the tissue samples, establishment of primary fibroblast cell lines, and chromosome preparation were performed as described before [9]. We tried to prepare fixed-cell suspensions for cytogenetic analysis at the earliest passages possible. A propensity for the emergence of tetraploid cells in pinniped fibroblast cultures has been noted previously [20]. We also observed a certain proportion of tetraploid cells, approximately 3–9%. This feature depended on the passage number, but the species karyotype was found to be stable in diploid cells.

### 2.3. Differential Staining

Standard GTG (G-bands by trypsin using Giemsa) banding staining was performed [21]. CBG (C-bands revealed by barium hydroxide treatment followed by Giemsa staining) [22] and CDAG (Chromomycin A3-DAPI after G-banding) methods [23] for CH visualization were used. To determine nucleotide composition by CDAG staining, two fluorochromes with opposite nucleotide specificity were utilized: DAPI (4′,6-diamidino-2-phenylindole; binds to adenine-and-thymine–rich regions; AT-binding) and CMA3 (Chromomycin A3; binds to guanine-and-cytosine–rich regions; GC-binding). Chromosomes in the karyotypes of all the analyzed pinniped species were arranged by length.

### 2.4. Preparation and Characterization of Chromosome-Specific Painting Probes

Sets of human, domestic dog, and stone marten chromosome–specific painting probes have been described previously [24,25,26]. In the present study, the dog chromosomal nomenclature follows to the one published by Yang et al. [24]. Whole-chromosome painting probe libraries of the domestic dog were employed for fluorescence in situ hybridization (FISH) analysis of genomes of one true seal species (the bearded seal *Erignathus barbatus*) and three eared seals (*Arctocephalus forsteri*, *Callorhinus ursinus*, and *Phocarctos hookeri*). Some human and stone marten painting probes were used to clarify ambiguous mapping positions in the ringed seal (*Phoca hispida*), spotted seal (*Phoca largha*), and harbor seal (*Phoca vitulina*).

### 2.5. Detection of Nucleolus Organizer Regions (NORs) and Telomeric Repeats

A plasmid containing ribosomal DNA (rDNA) [27] was amplified with the GenomePlex Whole Genome Amplification Kit (Sigma-Aldrich Co., St. Louis, MO, USA). Labeling of the plasmid DNA was performed by means of the GenomePlex WGA Reamplification Kit (Sigma-Aldrich Co.) with biotin-16-dUTP incorporation. Telomere repeats were synthesized and labeled by nontemplate PCR with primers (TTAGGG)_5_ and (CCCTAA)_5_ [28].

### 2.6. Image Acquisition and Data Processing

Digital images of hybridization signals were captured as described elsewhere [24,26,29] using the VideoTest system (Zenit, St. Petersburg, Russia) and a Zeiss microscope Axioscope 2 (Zeiss, Oberkochen,) equipped with a charge-coupled device (CCD) camera (Jenoptik, Jena, Germany). Images of metaphase spreads were edited in Corel Paint Shop Pro Photo X2 (Corel, Ottawa, Canada).

## 3. Results

### 3.1. Hybridization of Dog Probes onto Chromosomes of Four Pinniped Species

Dog (CFA) and some stone marten (MFO) chromosome-specific libraries were utilized to identify homologous segments in genomes of four pinniped species: one true seal and three eared seals. *Arctocephalus forsteri* and *Phocarctos hookeri* karyotypes are described here for the first time. Each CFA probe stained one to five fragments in the karyotypes of the bearded seal, Australian fur seal, northern fur seal, and New Zealand sea lion. Sixty-eight homologous autosomal segments were detected in the genomes of all the studied species. X-chromosome synteny was conserved among these pinniped karyotypes. Chromosome maps of the New Zealand fur seal, New Zealand sea lion, northern fur seal, and bearded seal are presented in Figure 1. Karyotypes of the pinnipeds analyzed here were deposited in the Atlas of Mammalian Chromosomes [4].

### 3.2. GTG Banding of Spotted Seal and Harbor Seal Chromosomes

We performed GTG staining of metaphase chromosomes of ringed and spotted seals. The *Phoca hispida* (2n = 32) chromosome set has been published previously [20]. The results on *Phoca largha* (2n = 32) presented here are consistent with the literature data [30]. The G-banding patterns of *P. hispida* and *P. largha* are very similar to the pattern in the Baikal seal (*Pusa sibirica*, 2n = 32; Figure 2 in ref. [9]).

### 3.3. Homologous Segments in Pinniped Genomes as Determined by FISH

The comparative chromosome painting helped us to identify homologous segments among the karyotypes of four other pinniped species in addition to previously published data on two true seals, the northern sea lion, and walrus [8,9] (see Appendix A).

An omission in our previous paper [9] is corrected here by the addition to the Ancestral Carnivora Karyotype (ACK) scheme the assignment of the fourth conserved element of CFA chromosome 28 onto ACK 9p dist. This additional fragment has not been seen in previous painting experiments on Carnivora [31,32]. In the Baikal seal, walrus, and northern sea lion, the comparative chromosome painting showed four fragments corresponding to CFA 28 (one on ACK 2, two on ACK 3, and one on ACK 9) [9]. In this study, we reliably detected the fourth fragment, homologous to chromosome CFA 28, in four other pinniped species: the bearded seal, New Zealand fur seal, northern fur seal, and New Zealand sea lion (Figure 1, chromosomes AFOR 7p, CURS 9p, PHOK 7p, and EBAR 9p).

### 3.4. CH in Pinniped Genomes

For identifying CH, both a standard banding cytogenetic method (CBG banding) and a new technique for the visualization of nucleotide composition of CH (CDAG method) were used as described in refs. [22,23]. Regions that did not hybridize with the painting probes in the FISH experiments are indicated by asterisks on the comparative chromosome maps (Figure 1) and were found to be composed of heterochromatin by CDAG consecutive G- and C-banding staining. The DAPI staining performed after the denaturation of the chromosome preparations subjected to the cross-species painting was taken into account too in this analysis. Here, we employed previously published comparative chromosome maps of the walrus and Steller sea lion [9], signifying areas of CH. These maps are given in the Appendix A.

In general, CH is scarce in genomes of pinnipeds. CBG staining yielded small pericentromeric blocks in walrus and true seal karyotypes except for the bearded seal (Figure 3a–d). Compared to true seals, the genomes of eared seals carry more CH (Figure 3f–i): all the studied eared seal species have additional telomeric heterochromatin segments on p-arms of some autosome pairs and larger pericentromeric blocks.

Using an array of staining assays, we determined the types, localization, and composition of heterochromatin areas in pinnipeds (Figure 4, Table 2). An interesting phenomenon was observed: parts of pericentromeric CH blocks were equally brightly stained by DAPI and Chromomycin A3. This was especially true for the bearded seal and less so for the walrus’s and eared seals’ chromosomes.

Our analysis of the alternation of dog painting probes’ staining patterns and in heterochromatin regions uncovered patterns of variation in CH among pinnipeds. Species chromosomal sets proved to differ in size and composition of pericentromeric and intercalary heterochromatin and in the number of autosomes carrying additional heterochromatin segments on short arms. Figure 5a–c shows the variation of the composition of homologous walrus and seals’ chromosomes corresponding to three ancestral autosomes: ACK 10 (MFO 11), ACK 11 (MFO 12), and ACK 5 (MFO 5).

### 3.5. The First Intrachromosomal Rearrangement Detected in Pinnipeds

The dog painting probes helped to identify for the first time an intrachromosomal rearrangement common for eared seals and the walrus (Figure 6). Among the pinnipeds, hybridization of the dog autosome 12 library revealed a difference in the localization of this probe relative to the centromere. In true seals, this conserved element is detectable as a whole near-centromeric segment on the long arm (EBAR 7q and PSIB 8q). In the walrus and eared seals, this segment is divided into two parts by the centromeric region, so that a small fragment of CFA 12 is detectable on the short arm of AFOR 8, CURS 7, EJUB 9, PHOK 8, and OROS 9. The chromosome level assemblies may be useful for testing whether an inversion or simple centromere repositioning has happened in an Otariidae/Odobenidae ancestor.

### 3.6. The Y-Chromosome Features of Pinnipeds

Male specimens were available for seven out of the 10 species included in this investigation. The Y chromosome is readily recognizable in the analyzed pinnipeds, being the smallest bi-armed chromosome of the complement (Figure 2 and Figure 3). Only in the northern sea lion, is the Y chromosome acrocentric and has a larger size, approximately half the length of Xq (Appendix A). As a rule, in seals, the CBG banding detected CH on the short arm and in the centromeric region of the Y chromosome. This part of the male gonosome must bear AT-enriched repeated sequences because it was DAPI positive. CMA3 highlighted the distal part of Yq in all the seals being analyzed (Figure 4). The male gonosome of the walrus differs from that of seals and contains CBG-, DAPI-, and CMA3-positive regions only in the centromeric part. We found an unusual NOR site on the Yp of the ringed seal *Phoca hispida*.

### 3.7. Localization of NORs and Telomeric Repeats

In all of the species studied here, the NOR was situated on one homologous autosomal pair (Figure 7). Ribosomal DNA sites unusual for pinnipeds were revealed on the Y chromosome of the ringed seal (*Phoca hispida*) and on autosome 6 of the harbor seal (*Phoca vitulina*). In both cases, these additional clusters of ribosomal genes were detectable in all of the analyzed cells and were substantially smaller than the main NOR site on PHIS 1p and PVIT 1p (s), respectively.

## 4. Discussion

The first and subsequent cytogenetic studies have detected low variation in the diploid number (32–36) and similar G-banding patterns among pinniped karyotypes [20,34,35,36], as confirmed later at the molecular cytogenetic level. Human and dog chromosome-painting probes have identified 22 and 68 conserved segments in the genomes of seals and walruses, respectively [8,9]. These findings reflect high conservatism of pinniped karyotypes and a low rate of their genome evolution in comparison with representatives of the canoid branch of Carnivora. More than twice as many human conserved autosomal segments have been found in highly rearranged genomes of Mephitidae (40), Ursidae (44), and Canidae (67–73) as compared to pinnipeds (22) [6,10,24,29,31,37,38,39]. In contrast to euchromatin conservatism, we found differences in the banding pattern and/or morphology of conserved syntenic groups across pinnipeds.

### 4.1. Heterochromatin Diversity among Pinniped Karyotypes

Conserved karyotypes of pinnipeds differ in the morphology of some autosomes shaped by varying heterochromatin regions. Heterochromatic regions are difficult to determine unless a special approach to the characterization of satellite DNA is used [40,41]. Regions of chromosomes enriched with repeated sequences are not highlighted during comparative painting because chromosome-specific painting probes contain libraries of unique sequences or evolutionarily distant repeats. Therefore, a combination of painting and different banding techniques should provide additional information about the euchromatin–heterochromatin structure of each chromosome.

We revealed an interesting phenomenon of composition diversity of centromeric and pericentromeric CH on homologous chromosomes among pinnipeds (Figure 5b,c, and Figure 6). Although most of the centromeres are GC-rich, there are autosomes with AT-enriched centromeric CH or those that do not show the predominance of certain nucleotide combinations. For example, in the bearded seal, some centromeres stained with equal intensity with both fluorochromes. Apparently, in such cases, AT- and GC-enriched repeats are interspersed in the centromeric regions.

We observed variation of p-arms’ heterochromatin (Figure 5a–c and Figure 6) after detecting an addition of the heterochromatin blocks and/or expansion of a heterochromatin block, in an extreme case of CURS11p, causing an increase in the fundamental number (FNa = 70) as compared to the northern sea lion (FNa = 68) [20,34]. We also observed rare intercalary heterochromatin sites in CURS4q and CURS10q (Figure 5c). These CH additions and expansions do not change the order of syntenic segments. In several cases, conserved synteny was found to be split into two parts owing to heterochromatic near-centromeric inserts as in eared seals and the walrus (Figure 5b and Figure 6).

Heterochromatin segments may be conserved too, as in apomorphy. One such instance inferred from comparative chromosome maps is the chromosomes homologous to cat autosome Felis catus, CFA) A1q (HSA 5). This chromosomal segment carries an additional heterochromatic short arm in studied eared seal (except for AFOR 5p) and in other carnivores’ karyotypes analyzed by chromosome painting, indicating an ancestral origin of this heterochromatin segment [26,31,32,42,43]. Homologous autosomes with similar heterochromatin patterns belong to different pinniped branches (Figure 5a), and this effect can be regarded as an example of plesiomorphy.

In summary, the majority of ancestral syntenies showed a wide variety of CH characteristics (localization, size, and nucleotide composition) despite small sizes of heterochromatin areas. Among the studied species, the numbers of autosomes affected by CH variation were different. Except for mustelids, the corresponding outgroup homologs in the stone marten, felids, and other carnivoran taxa do not feature such CH variation [10,32]. The heterochromatin diversity is also expressed in the variation in the NOR and sex chromosomes among some pinniped species (SM1 and SM2). In toto, we revealed a small amount of CH in the pinniped genomes, with the largest amount found in the northern fur seal, walrus, and bearded seal and the smallest amount in the Baikal seal. Overall, our findings suggest that the variation in heterochromatin has played a substantial role in pinniped karyotype evolution.

Karyotypes of all pinnipeds are similar, but at the same time, the species living together seem to retain reproductive isolation. For example, there are no cytogenetic reports about interspecific hybrids of three sympatric species: the ringed seal, spotted seal, and bearded seal. Interspecific hybrids of southern fur seals of the genus *Arctocephalus* are known, but hybrid males have lower reproductive success [44]. It is possible that the differences in heterochromatin elements affect the pairing of homologous chromosomes during hybrid meiosis and diminish the formation of viable gametes.

The involvement of heterochromatin in genome functioning, adaptation, and speciation is still a fundamental problem of modern genetics. The notion that CH is a genetically inactive part of the genome has been refuted [1,45,46,47,48,49,50]. The position effect on gene expression depends on the proximity to the site of CH and apparently may reflect intranuclear location of genes at interphase [51,52,53,54]. The observed interspecies diversity among pinnipeds suggests that the combinations of hetero- and euchromatin segments observed in the karyotypes of different species may be adaptive. Heterochromatin regions in pinnipeds may contribute to interspecific differences and may perform a regulatory function. Given that species of fur seals differ in forage and sexual behavior, in characteristics of the fur coat, and in vocalization [55,56,57], further research is needed to determine functional properties of the repeated sequences in their genomes.

### 4.2. The Intrachromosomal Rearrangement Detected in Pinnipeds

We noticed differences in the distribution of CFA 12 on p- and q-arms in homologs of ancestral synteny ACK 6. In true seals, CFA 12 stained only the proximal part of the q-arm, but in the walrus and eared seals, this probe additionally highlighted a proximal small part of the p-arm (Figure 6). This finding may be the result of a pericentric inversion or centromere repositioning.

The application of high-resolution painting probes of the domestic dog and raccoon dog has revealed cryptic inversions both in Caniformia and Feliformia branches of Carnivores [10,32,37,39,42,58,59,60]. It has been reported that some ancestral autosomes are more prone to inversions in different families and that some of these rearrangements may represent valuable cytogenetic signatures [32]. Inversion 37/**12/1/12/1** in conserved synteny CFA 37/12/1 is common for at least three mustelid species (the stone marten [32], American mink [58], and European badger (our unpublished data)) and involves breaks in two conserved segments (CFA 1 and CFA 12). The rearrangement in the same syntenic element in pinnipeds implies changes only in CFA 12. Consequently, different transformations have occurred in the same syntenic element in different Carnivora groups having conserved genomes.

An alternative explanation for the observed difference in the structure of the ACK 6 homolog revealed in both Otariidae and Odobenidae is a centromere repositioning without a change in the gene order. New centromeres (neocentromeres) and centromere shifts have been detected in various vertebrate species [61] including humans [62,63]. The sites commonly affected by the centromere movements revealed in primates are located on the long arm of ACK 6, whereas in pinnipeds, the supposed centromere displacement is directed toward the short arm (Figure 6). The nature of the intrachromosomal rearrangement on ACK 6 homologs in pinnipeds (Appendix A) may be determined by the mapping of region-specific painting probes or BAC clones.

A potential rearrangement was detected here in the homologous segment on EBAR 10: painting probe CFA 11 highlighted a twofold larger portion of the chromosome than that in eared seals (Figure 6), likely indicating a duplication. The polyploidization of primary cell lines reported earlier [20] in pinnipeds remains unexplained and may be used in the future to study the mechanisms of genome instability.

### 4.3. The ACK Validated by the New Pinniped Painting Data

Reconstruction of ancestral karyotypes for taxa at different levels is an important step for the compilation and classification of comparative cytogenetic data. The hypothesis about karyotype structure of the ancestor of all Carnivora has been corrected at each stage of cytogenetics development. At first, a “hypothetical primitive karyotype” with at least 2n = 34 [64] was published. Then, an ancestral karyotype called CAR containing 21–22 autosome pairs was proposed, taking into account gene-mapping results on human and domestic cat [65,66]. A derivative hypothesis about ACK composition is based on Zoo-FISH data from the cat and harbor seal [8]. This Z-CAR karyotype consists of 38 elements, as is the ACK reconstructed by comparative painting of Feliformia and Caniformia [10].

Being a basal canoid group, pinnipeds are an important lineage for ACK reconstruction if canoid families with substantially reorganized karyotypes (canids, ursids, and skunks) are ignored. Consequently, the new chromosome maps of pinnipeds confirm 2n = 38 for the ACK. Elements ACK3p/3q and ACK4p/4q are fused thereby reducing 2n from 42 to 38 [5]. The ACK syntenic groups with the order of conserved segments updated with CFA28d, are presented in Figure 8. Overall, the ACK is now fully integrated with the available painting data on key reference genomes (HSA, CFA, FCA, and MFO; Appendix A).

### 4.4. Chromosome Evolution in Pinnipedia

The ancestral pinniped karyotype is identical to the ACK because no rearrangements have been identified on a branch leading from the ACK to Pinnipedia [9]. The radiation into pinniped families has been accompanied by several fusions leading to branches of Otariidae + Odobenidae, Phocidae, Phocini, and Odobenus. ACK chromosomes 5, 16, and 18 were repeatedly involved in fusions in pinnipeds, and the fusion sites include the same chromosomal termini likely prone to fusion. In two fusions, the formation of new autosomes was accompanied by the appearance of evolutionarily new centromeres [61,62] (ACK 12/16 in the Phocidae branch and ACK 16/18 in Odobenidae + Otariidae; Figure 9). We identified one inversion/centromere repositioning that took place during the formation of Phocidae/Odobenidae + Otariidae genomes. Specific accumulation of heterochromatin components has occurred in each group of pinnipeds after separation into an independent branch (Figure 9, designated +H).

Recently, chromosome level genome assemblies for several pinniped species were released: for *Odobenus rosmarus* (2n = 32)*, Phoca vitulina* (2n = 32)*, Erignathus barbatus* (2n = 34)*, Neomonachus schauinslandi* (monk seal, 2n = 34)*,* and *Mirounga angustirostris* (northern elephant seal, 2n = 34) [19]. The monk seal and northern elephant seal belong to the Monachinae subfamily with sister branch Phocinae (*Erignathus*, *Phoca*, and *Pusa* here) inside Phocidae. There were no fusion/fission events during Monachinae formation as evidenced by pairwise comparative plots of chromosome level scaffolds between these species and *E. barbatus*.

The first cytogenetic studies on marine mammals in the 1970s–1980s revealed the conservatism of chromosome sets and the stability of diploid numbers: 2n = 44 for most Cetacea and 2n = 32 to 36 in pinnipeds [12]. The observed karyotype uniformity is hypothesized to derive from characteristics of the habitat and physiology. Comparative chromosome painting has uncovered an identical order of syntenic segments in toothed and baleen whales [32,73,74] as well as in delphinids [75], just as among the pinnipeds in the present study. The low numbers of interchromosomal rearrangements giving rise to high conservatism of syntenic segments in cetacean and pinniped marine mammals seem well established.

In addition, the evolution of the variable heterochromatic part of the genome in different groups of marine mammals has been researched actively in whales and less so in pinnipeds. Heterochromatin in whales reaches 25–30% of the genome and is composed of different families of repeated sequences [76,77,78,79,80,81,82]. Heterochromatin is scarcer in pinnipeds than in whales. The walrus and the studied species of seals display their own distinctive patterns of heterochromatin regions and underlying repeat family diversity. Precise composition and organization of repeated sequences in pinniped genomes remain to be determined. It seems clear, however, that molecular genetic approaches have proved the second general property of marine mammals’ genomes: CH variation against the background of high conservatism of the euchromatin part.

## 5. Conclusions

Here we expanded to eight the list of pinniped species with comparative chromosomal maps. The resulting data confirm the high conservatism of syntenic elements in this group of marine mammals relative to other carnivore groups. We postulate that the order of evolutionarily conserved segments is common among the analyzed pinnipeds and is the same as that in the proposed karyotype of the ancestor of the whole order Carnivora. Fusions of ancestral-karyotype elements are specific to each branch of pinnipeds. Evolutionary changes at the subchromosomal level were revealed in two events of evolutionarily new centromeres and one pericentric inversion/centromere repositioning. The interspecific diversity due to variations in the heterochromatin component of pinniped karyotypes are described for the first time. Thus, the karyotypic divergence of conserved genomes of pinnipeds has occurred largely at the level of heterochromatin or via intrachromosomal repositioning.

## Figures and Tables

**Figure 1 genes-11-01485-f001:**
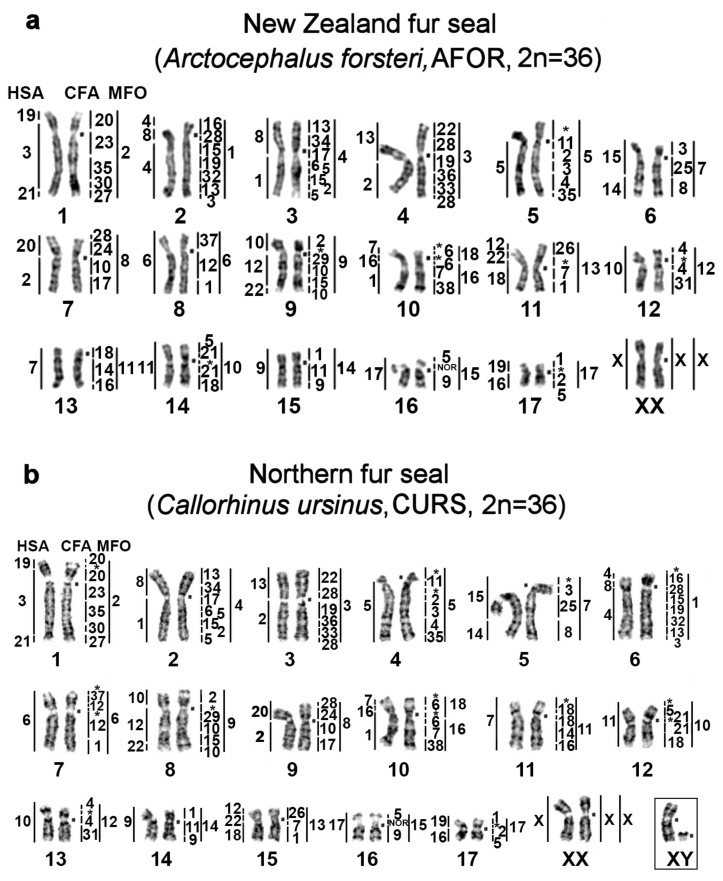
GTG banded karyotypes of (**a**) the New Zealand fur seal (AFOR, 2n = 36), (**b**) northern fur seal (CURS, 2n = 36), (**c**) New Zealand sea lion (PHOK, 2n = 36), and (**d**) bearded seal (EBAR, 2n = 34) with the assignment of homology to human (HSA), dog (CFA), and stone marten (MFO) chromosomes. Nucleolus organizer regions are marked as NOR. The square denotes a centromere position on the corresponding chromosome. * Heterochromatin regions not painted by any dog probe.

**Figure 2 genes-11-01485-f002:**
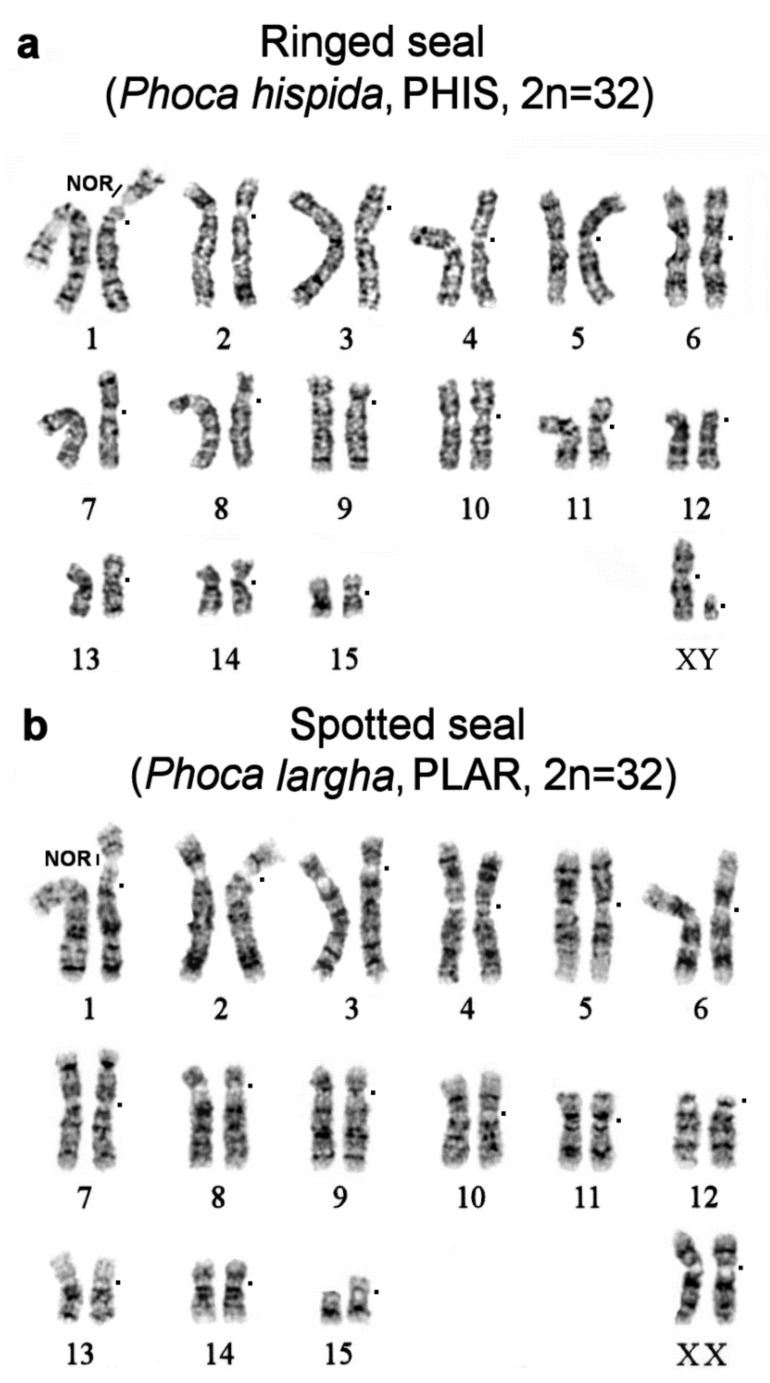
The GTG banded karyotype of (**a**) the ringed seal (PHIS, 2n = 32) and (**b**) spotted seal (PLAR, 2n = 32). The square denotes a centromere position. Sites of rDNA gene clusters are marked as NOR.

**Figure 3 genes-11-01485-f003:**
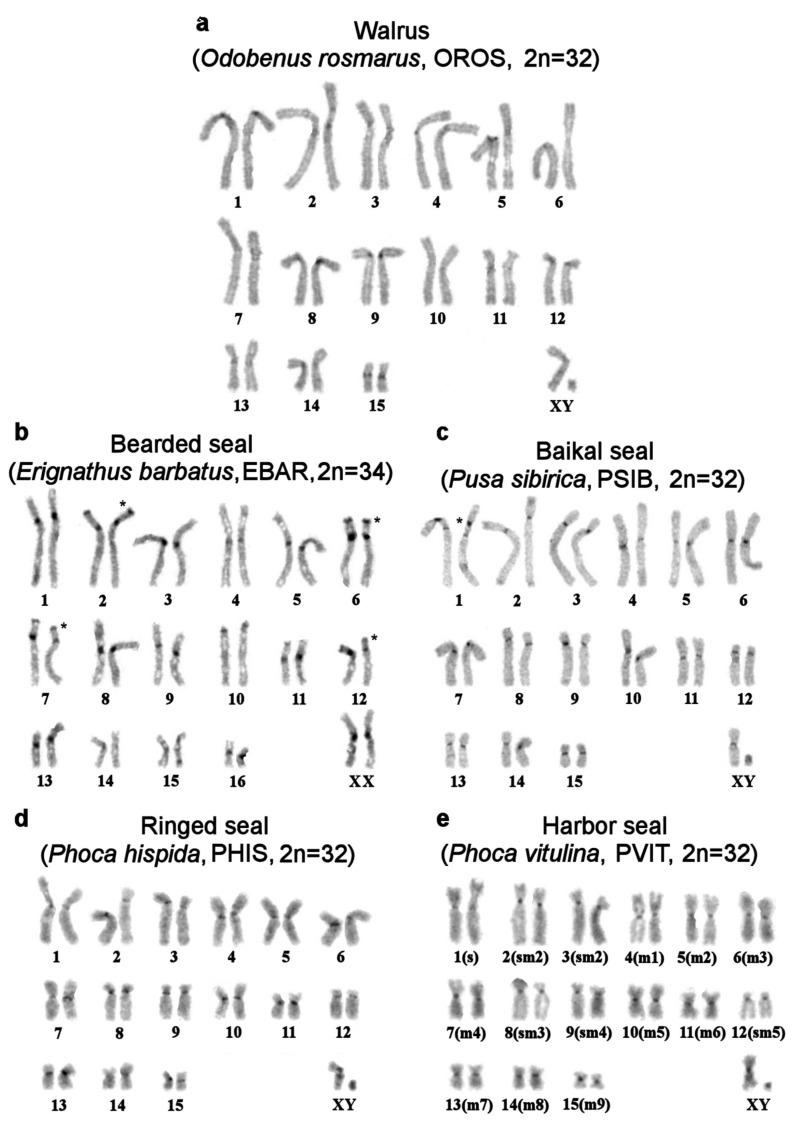
CBG banded karyotypes of (**a**) the walrus and true seals: (**b**) the bearded seal *, (**c**) Baikal seal **, (**d**) ringed seal, and (**e**) harbor seal. For the harbor seal, as is the case for all the species in this article, chromosomes are arranged in descending order. Correspondence to the harbor seal nomenclature of Fronicke et al [8] is given in parentheses. * The C-banding fully matches the one published earlier [20]. Note additional telomeric heterochromatin segments on p-arms of four autosome pairs in EBAR (asterisk). ** Note an intercalary heterochromatin segment in PSIB1p near the region containing rDNA genes (asterisk). CBG banded karyotypes of the eared seals: (**f**) the New Zealand fur seal, (**g**) northern fur seal *, (**h**) New Zealand sea lion, and (**i**) northern sea lion. * Interstitial heterochromatin in CURS1p, CURS 4q, and CURS10q and additional telomeric heterochromatin segments on p-arms are marked by an asterisk.

**Figure 4 genes-11-01485-f004:**
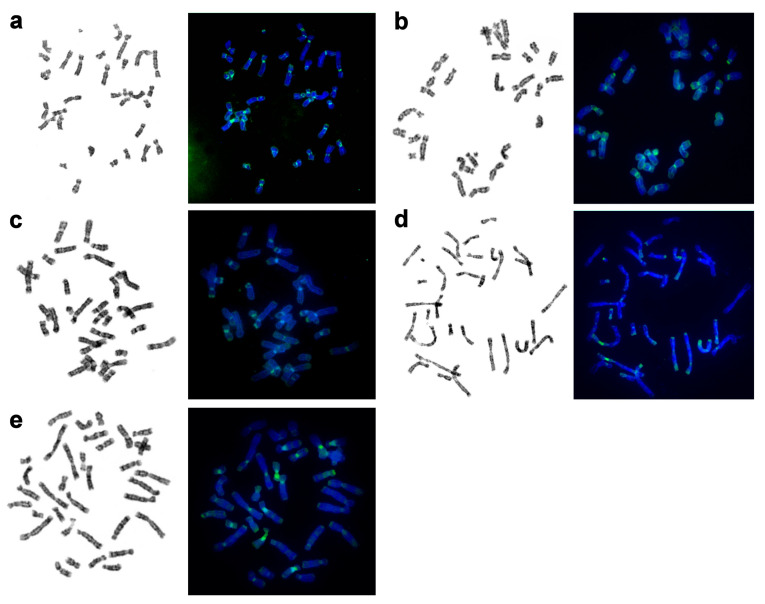
CDAG staining of metaphase spreads of pinnipeds: GTG banding (left) followed by formamide heat denaturation and CMA3/DAPI staining (right). (**a**) The walrus (OROS), (**b**) bearded seal (EBAR), (**c**) ringed seal (PHIS), (**d**) Baikal seal (PSIB), and (**e**) northern sea lion (EJUB).

**Figure 5 genes-11-01485-f005:**
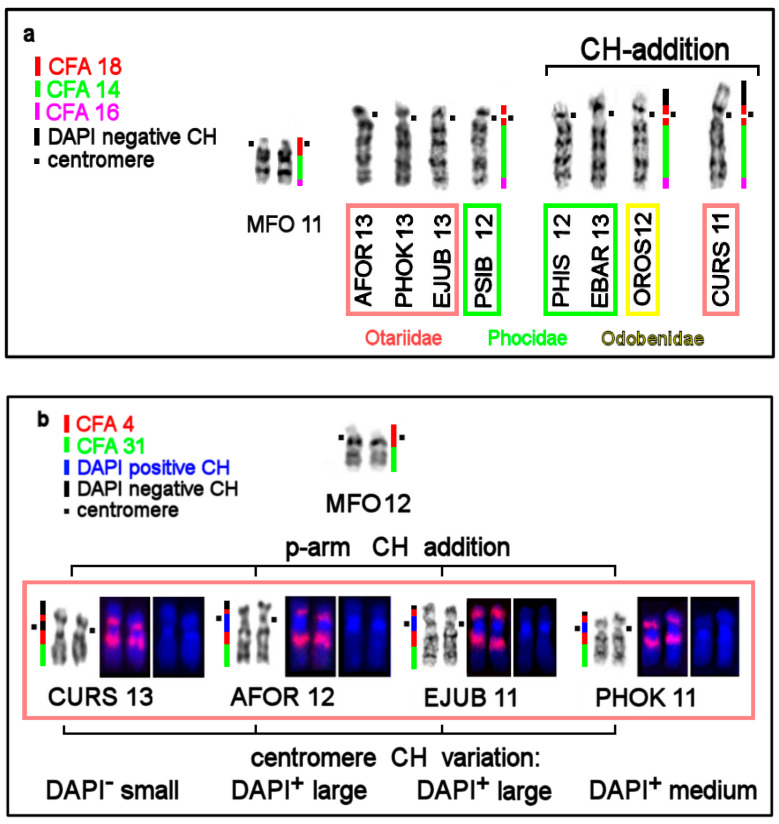
The variation in the amount, composition, and distribution of heterochromatin in conserved syntenic groups of pinnipeds. Fluorescent in situ hybridization of CFA11 (green) and CFA2 (red) on DAPI (blue) stained chromosomes. (**a**) CH addition in four species of pinnipeds on p-arms of homologs of stone marten chromosome MFO11. (**b**) CH addition on p-arms, centromere size, and nucleotide composition variation on MFO 12 homologs in eared seals. (**c**) Different sizes and nucleotide compositions of heterochromatin p-arms among the eared seals. Intercalary heterochromatin in CURS 4q is DAPI positive. Note the possible duplication of the segment homologous to CFA 11 in EBAR10 and PSIB1. The segment order was revealed by G-banding followed by mapping of dog (CFA) painting probes (red and green in FISH images of DAPI-stained chromosomes). Segments that did not stain with any dog probe represent CH and are highlighted in blue (DAPI^+^) or black (DAPI^−^). Stone marten (MFO) chromosomes representing the ancestral form of the syntenic group in Carnivora are borrowed from ref. [32]. Pinniped families are designated by color boxes: pink, Otariidae; green, Phocidae; and yellow, Odobenidae.

**Figure 6 genes-11-01485-f006:**
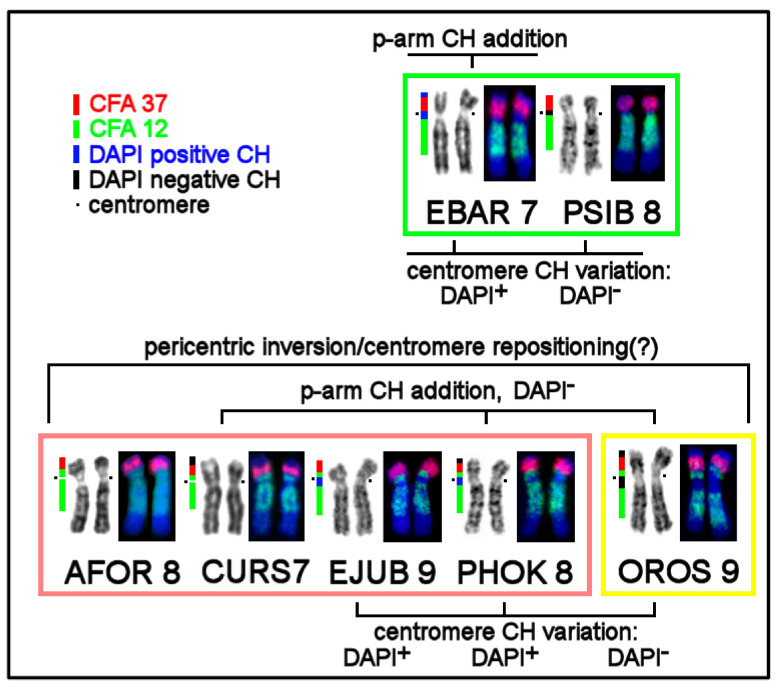
Pericentric inversion or centromere repositioning (?) in Odobenidae and Otariidae as revealed by dog painting probes CFA 37/12 (ACK 6). Pinniped families are designated by color boxes: green, Phocidae; pink, Otariidae; and yellow, Odobenidae.

**Figure 7 genes-11-01485-f007:**
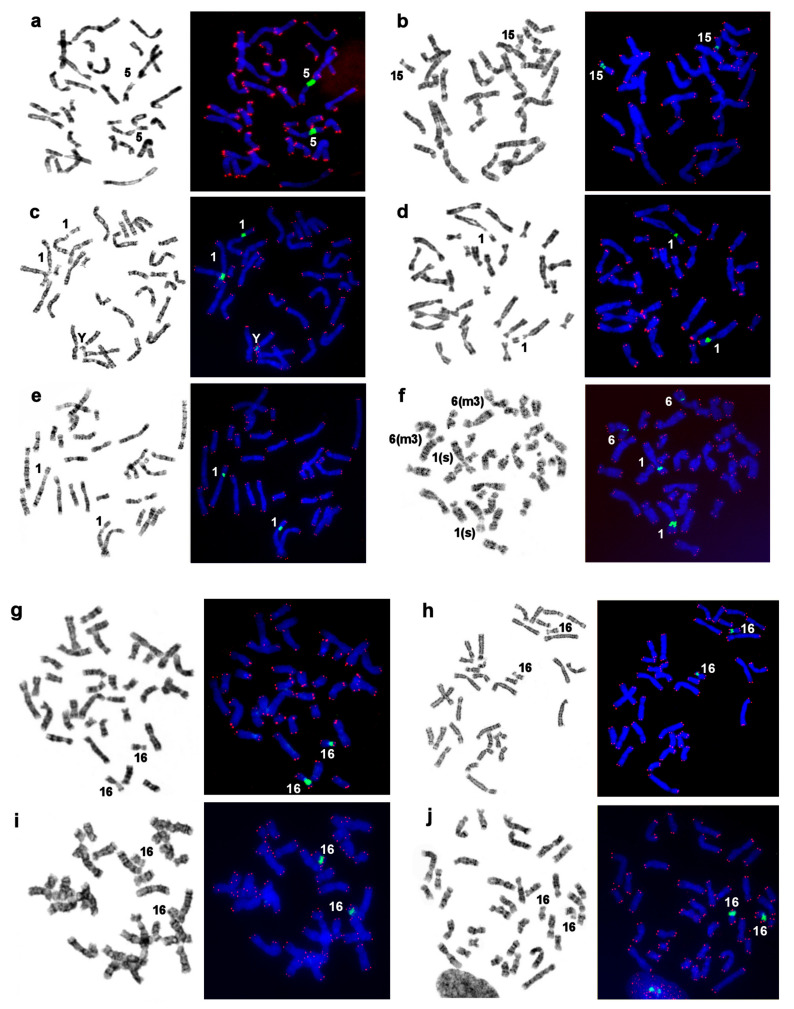
Distributions of telomeric repeats and rDNA clusters on chromosomes of pinnipeds. G-banding (left) followed by FISH (right) of a telomeric probe (TTAGGG)_n_ (red) and an 18S, 5.8S, and 28S rDNA probe (green) on DAPI-stained chromosomes. (**a**) The walrus and true seals, (**b**) the bearded seal, (**c**) ringed seal, (**d**) spotted seal, (**e**) Baikal seal, and (**f**) harbor seal. The telomeric probe highlighted the termini of all chromosomes in the investigated species. Note differences in the size and signal intensity on telomeres (some were very faint) in species, with the walrus’s telomeric blocks being larger than those in the other species. No interstitial telomeric sites marking fusions of ancestral elements were detectable in pinnipeds here [33]. The chromosome numbers are indicated for NOR bearing chromosomes. NOR sizes vary between homologs among several species. The eared seals: (**g**) the New Zealand fur seal, (**h**) northern fur seal, (**i**) New Zealand sea lion, and (**j**) northern sea lion.

**Figure 8 genes-11-01485-f008:**
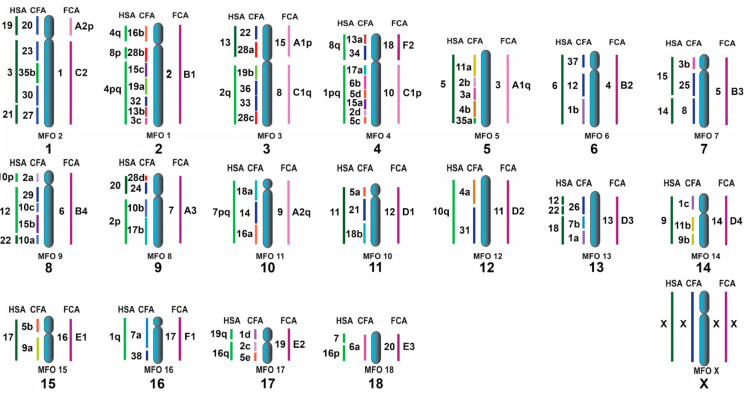
The ACK (2n = 38) updated according to [9]. Conserved and rearranged chromosomes in each species (FCA and HSA) are highlighted in different shades (dark and light). In CFA, the chromosomes represented by a single element in the ACK are shown in dark blue, whereas split chromosomes are each highlighted in a different color. To the right of the ancestral chromosomes is the number from the ACK 2n = 42 nomenclature [5]. All seven syntenic associations of human chromosomes shared by representatives of the Eutheria clade (HSA 3/21, 4/8p, 7/16p, 10p/12pq/22qt, 12qt/22q, 14/15, and 16q/19q) [5,6,7,37,67] were detected in pinniped species [8,9]. Our present work confirmed that one fission (HSA1), one inversion (HSA4q/8p/4pq), and five fusions (1q/8p, 2q/13, 2/20, 19p+3/21, and 12qt/22pq+18) distinguish the ACK from the ancestral Eutherian karyotype [6,7,8,9,10].

**Figure 9 genes-11-01485-f009:**
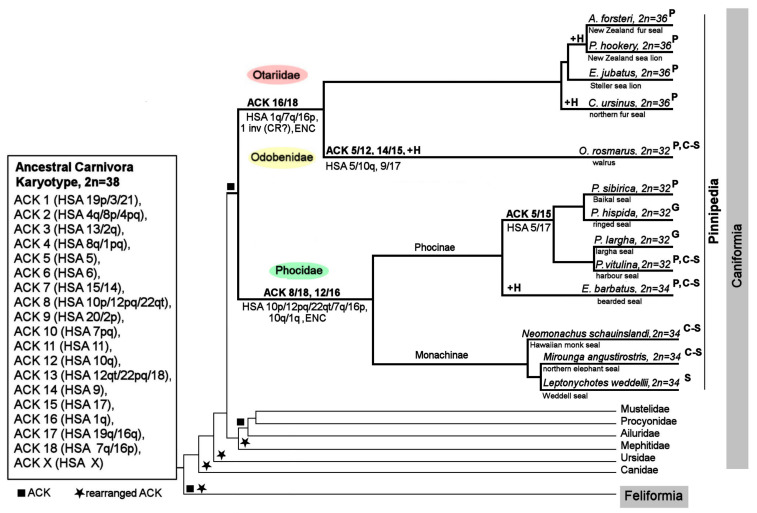
A compiled scheme of karyotype evolution in Pinnipedia, including the species studied here and species for which whole-genome sequencing data are available. The fusion of ancestral elements is designated by “/”. +H: heterochromatin addition, ENC: evolutionarily new centromere, inv (CR?): inversion or centromere repositioning. ^G^ G-banding only, ^P^ painting data, ^S^ whole-genome sequencing data, and ^C-S^ chromosome level genome assembly (C-Scaffolds) [19,68]. The cytogenetic data are consistent with the tree topology based on published Carnivora phylogenies affirming pinniped monophyly [69]. The basal position of the walrus is supported by two extra fusion events [70]. The lineage of the northern fur seal first split from Otariidae ~8 mya [71,72]. The separation of the bearded seal ~12 mya from the rest of Phocinae [70] is confirmed here. Substantial enrichment with repeated sequences with variation of nucleotide composition has occurred during long-term monotypic evolution in *O. rosmarus*, *C. ursinus*, and *E. barbatus* in contrast to fewer heterochromatin changes in evolutionarily recent pinniped species. Cytogenetically, *Phoca* and *Pusa* and likely other genera with 2n = 32 on this branch are linked by a single fusion (ACK 5/15). Overall, pinniped karyotype evolution has had a slow rate of genome rearrangements with less than one rearrangement per 10 million years [5].

**Table 1 genes-11-01485-t001:** Description of the species used in this study.

No	Family	Latin Names	Code	2n	Sex	Common Names	Reference for Fluorescence In Situ Hybridization (FISH) Data
1	Odobenidae Walruses	*Odobenus rosmarus*	OROS	32	M	walrus	[9]
2	Otariidae Eared seals	*Arctocephalus forsteri*	AFOR	36	F	New Zealand fur seal (South Australian fur seal)	this article
3		*Callorhinus ursinus*	CURS	36	M	northern fur seal	this article
4		*Phocarctos hookeri*	PHOK	36	M	New Zealand sea lion	this article
5		*Eumetopias jubatus*	EJUB	36	M	northern sea lion (Steller’s sea lion)	[9]
6	Phocidae True seals	*Erignathus barbatus*	EBAR	34	F	bearded seal (square flipper seal)	this article
7		*Phoca hispida*	PHIS	32	M	ringed seal	
8		*Phoca largha*	PLAR	32	F	spotted seal (largha seal)	
9		*Phoca vitulina*	PVIT	32	M	harbor seal (common seal)	[8]
10		*Pusa sibirica*	PSIB	32	M	Baikal seal	[9]
11	Canidae	*Canis familiaris*	CFA	78		domestic dog	
12	Mustelidae	*Martes foina*	MFO	38		stone marten	
13	Hominidae	*Homo sapiens*	HSA	46		human	

**Table 2 genes-11-01485-t002:** Distribution and nucleotide composition of the heterochromatin blocks in pinnipeds.

Location of CH	AT/GC Composition of CH Detected by Fluorochromes DAPI and CMA3	Notes
**Centromeric region**	varies but GC-rich in most species: CMA3^+^	
	DAPI^+^	near-centromeric blocks on some autosomes of walrus and true seals (but not in bearded seal
	CMA3^+^/DAPI^+^	4–7 small and mid-sized autosomal pairs in eared seals
**Pericentromeric region in q-arms**	AT-rich: DAPI^+^	
**Terminal CH in p-arms**	GC-rich in most species: CMA3^+^; DAPI^+^ on AFOR 5p	absent in *Phoca* and *Pusa*
**Interstitial CH**	AT-rich: DAPI^+^	in eared seals

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
