# Peer review of "Karyotype Evolution in 10 Pinniped Species: Variability of Heterochromatin versus High Conservatism of Euchromatin as Revealed by Comparative Molecular Cytogenetics"

_genes, 2020, doi:10.3390/genes11121485_

Round 1

Reviewer 1 Report

The manuscript brings valuable data on karyotypes in 10 pinniped species including comparative analysis using painting probes from three mammal species and I believe it deserves to be published. I have no comments on the analyses and result interpretations, I believe it has been done very well. I found it, however, difficult to read, as a person who is not familiar with Pinnipedia. Thus, I have some suggestions which can make this article more user-friendly.

1) Please do not switch between Latin and common names in the text, it is difficult to follow which common name belongs to which Latin name.

2) Please indicate somewhere (e.g. Tab. 1 and/or Fig. 9) which are the true and eared seals.

3) You mentioned that karyotypes of Phoca hispida and P. largha based on G-banding were already published. Is there any purpose for figure 2 which presents these karyotypes again? Maybe it could be moved to the supplements.

4) Karyotypes in Figure 3 are too small, please make them larger. Perhaps indicate the heterochromatin segments in p-arms mentioned in line 187 with arrows. Why C-banding was not done on Phoca larca

5) Figure 4 is too small. Also, it would make the hybridization pattern more clear if the green and blue channels were presented separately or blue turned to grey or green changed to red. Green on blue in the small picture makes the small blocks nearly invisible. The same applies to fig. 7 - small rDNA clusters are very hard to see.

6) In figure 6 the green on blue is hard to see, would be better to present blue and green separately or turn DAPI to grey or switch red and green (as the green hybridization is more discussed in the text).

7) Lines 249 and 250: There is this sentence "Only in the Northern sea lion the Y chromosome is acrocentric and has a larger size ..." but there is no picture showing it in the manuscript nor a reference.  

8) Line 409 - To be consistent, add chromosome numbers to all species. Similarly, explain all abbreviations used in supplementary table 1.

9) Would be nice to add names of subfamilies to the cladogram in fig. 9., as the are mentioned in the text.

10) Line 424 I suppose there should be Phocinae instead of Phocidae.

11) Line 137 - I think the Australian fur seal is in fact the New Zealand fur seal. Similarly, the common seal mentioned at the end of the second paragraph of Supplementary material 1 is probably the harbor seal.

12) There is no legend in supplementary figure 1.

Author Response

Reply review report

“Karyotype evolution in ten pinniped species: variability of the heterochromatin versus high conservatism of the euchromatin revealed by comparative molecular cytogenetics”

Reviewer 1

1) Please do not switch between Latin and common names in the text, it is difficult to follow which common name belongs to which Latin name.

Response 1: Thank you for this suggestion. We removed species Latin names from the text and replaced them with the common names. 

2) Please indicate somewhere (e.g. Tab. 1 and/or Fig. 9) which are the true and eared seals.

Response 2: Thank you for this suggestion. We added this information to Table 1. We also filled the rows in the table with different colors for different families: green, Phocidae; pink, Otariidae; and yellow, Odobenidae

3) You mentioned that karyotypes of Phoca hispida and P. largha based on G-banding were already published. Is there any purpose for figure 2 which presents these karyotypes again? Maybe it could be moved to the supplements.

Response 3: Karyotypes of these species were submitted to the Atlas of Mammalian Chromosomes (2020) prior to revealing clusters of ribosomal genes (NORs). So we included karyotypes of P.hispida and P.largha in the Manuscript with updated information. Also, the Atlas of Mammalian Chromosomes is not an open-access and we would like to keep these two karyotypes in the body of the paper to show the conservatism of pinniped karyotypes.

4) Karyotypes in Figure 3 are too small, please make them larger. Perhaps indicate the heterochromatin segments in p-arms mentioned in line 187 with arrows. Why C-banding was not done on Phoca larca

Response 4: Thank you for this suggestion. Heterochromatin segments in p-arms and also interstitial-heterochromatin regions are shown by asterisks. We are going to continue this research on pinniped karyotypes. CBG-banded chromosomes of Phoca largha will be included in the next manuscript. 

5) Figure 4 is too small. Also, it would make the hybridization pattern more clear if the green and blue channels were presented separately or blue turned to grey or green changed to red. Green on blue in the small picture makes the small blocks nearly invisible. The same applies to fig. 7 - small rDNA clusters are very hard to see.

Response 5: Thank you for the comment about image perception. We worked to improve images. We would like to keep green and blue colors for the CDAG-staining (Fig.4) as been described in the original publication (Lemskaya et al., 2018). We have made pictures brighter attempting to make it easier to see the signal for the reader. Oftentimes the CDAG signal has subtle shades indicating combined presence of AT and GC-rich areas. These subtle details would be lost while contrasting or converting the picture to black and white.

We also increased the size of Fig.1 -9. We added numbers for chromosomes bearing clusters of ribosomal genes to the pictures with hybridization results on Fig. 7.

6) In figure 6 the green on blue is hard to see, would be better to present blue and green separately or turn DAPI to grey or switch red and green (as the green hybridization is more discussed in the text).

Response 6: Thank you for the comment about image quality and perception. We have changed the image to make it more clear by increasing the brightness. The  DAPI+ and DAPI- segments are clearly seen on the DAPI-only figures (like on Figure 5 on the right). However, we did not include these additional DAPI-only images onto this figure depicting DAPI+ and DAPI- segments on the color schemes on the left. In this picture we really wanted the readers to focus on the inversion/centromere reposition.

7) Lines 249 and 250: There is this sentence "Only in the Northern sea lion the Y chromosome is acrocentric and has a larger size ..." but there is no picture showing it in the manuscript nor a reference.

Response 7: We added the reference to the Figure S1a.

8) Line 409 - To be consistent, add chromosome numbers to all species. Similarly, explain all abbreviations used in supplementary table 1.

Response 8: We added chromosome numbers and abbreviations..

9) Would be nice to add names of subfamilies to the cladogram in fig. 9., as they are mentioned in the text.

Response 9: Phocidae subfamilies were added to the cladogram on Fig.9.

10) Line 424 I suppose there should be Phocinae instead of Phocidae.

Response 10: Thank you. We corrected it. 

11) Line 137 - I think the Australian fur seal is in fact the New Zealand fur seal. Similarly, the common seal mentioned at the end of the second paragraph of Supplementary material 1 is probably the harbor seal.

Response 11: We added alternative common names of pinniped species to Table 1. Yes, the New Zealand fur seal is in fact the South Australian fur seal.

12) There is no legend in supplementary figure 1.

Response 12: We added the legend to figure 1.

Reviewer 2 Report

The MS presents some detailed comparative data on the karyotypes of 10 pinniped species, including several species where data are presented for the first time.

The study is well executed and the data appropriately analyses and interpretted.

However the MS needs some improvement

  1. the quality of written English needs to be improved throughout
  2.  consistency is required in the use of common names, scientific names and species codes throughout. Currently it is very hard to follow are all are used interchangeably. i would suggest giving the common name, followed by the scientific name and species code  when a species is first mentioned and then just consistently using one of these throughout.
  3.  scientific names need to be italicised throughout
  4.  lines 37-39. rephrase as meaning unclear
  5. line 65 not clear what 'versus' means in this context
  6.  Lines 86-98 needs to be condensed.

Author Response

Reply review report

“Karyotype evolution in ten pinniped species: variability of the heterochromatin versus high conservatism of the euchromatin revealed by comparative molecular cytogenetics”

Reviewer 2

1) The quality of written English needs to be improved throughout

Response 1: We used the help of a professional English editor to improve the English in the text.

2) Consistency is required in the use of common names, scientific names and species codes throughout. Currently it is very hard to follow are all are used interchangeably. i would suggest giving the common name, followed by the scientific name and species code  when a species is first mentioned and then just consistently using one of these throughout.

Response 2: Thank you for this comment. We removed the Latin names of the species from the text replacing them with the common names. We also added synonyms of the common names of pinniped species to Table 1.

3) Scientific names need to be italicised throughout

Response 3: Thank you for the comment. We checked the text to italicize all Latin names.

4) Lines 37-39. Rephrase as meaning unclear

Response 4: Thank you for the comment. We have edited the sentence for clarity:

The observed interspecific diversity of pinniped karyotypes driven by constitutive heterochromatin variation likely has played an important role in karyotype evolution of pinnipeds thereby contributing to the interspecific differences of pinnipeds’ chromosome sets.

5) Line 65 not clear what 'versus' means in this context

Response 5: Thank you for the comment. We have edited the sentence for clarity:

Carnivora (includes two branches Feliformia and Caniformia)  is one order with well-characterized patterns of chromosome evolution [11]. There are two types of evolutionary changes that dominate in certain groups of the order: chromosomal sets of Canidae, Ursidae, and Viverridae were derived mainly through the fusion-fission of ancestral elements whereas inversions, centromere repositions have been mostly detected during karyotype evolution of  other families of Carnivora .

 6) Lines 86-98 needs to be condensed.

Response 6: Thank you for your comment. We have shortened the paragraph.
